# Application of a dynamic optimization-based multi-attribute fusion method for fault detection

Chen Ma[1¤], Handong Huang[1¤*], Youcai Tang[1], Suo Cheng[2], Chao Wang[2], Xin Wang[2]

1 State Key Laboratory of Petroleum Resources and Prospecting, College of Geophysics, China University of Petroleum, Beijing, China, 2 Exploration and Development Research Institute, PetroChina Tarim Oilfield Branch, Korla, China

¤ College of Geophysics, China University of Petroleum, Fuxue Road, Changping District, Beijing, China
* webhhd@163.com, tangyc@cup.edu.cn

## Abstract

The focus of oil and gas exploration in the Tarim Basin has shifted from interlayers to fracture-controlled karsts. A significant oilfield characterized by strike-slip faults was discovered in the Yueman area. However, identifying such fault zones is challenging because of the complex and chaotic seismic reflection characteristics, as well as the presence of seismic noise and other discontinuities. To improve oilfield production, the accurate identification of strike-slip fault zones in ultradeep tight limestone is a critical issue. The seismic anomalies of such fault zones exhibit diverse characteristics, with the low-velocity zone of the fault causing a "beaded" reflection pattern. Traditional coherent and curvature attribute methods have large errors in identifying strike-slip faults and cannot adequately characterize the contour features of fracture-karst traps. To address these challenges, this study proposed a multi-attribute optimal surface-based fracture identification technology based on forward simulation records. Seismic attributes that were sensitive to different types of strike-slip faults were selected, and multiple attributes were merged to obtain a fracture distribution map using the best surface voting algorithm. This method effectively suppresses noise that is irrelevant to fractures and is sensitive only to fracture information, allowing for the identification of subtle waveform changes caused by strike-slip faults. Thus, the accuracy and continuity of fracture identification were significantly improved.

## Introduction

Deep-seated strike-slip faults in western China play a pivotal role in controlling substantial oil and gas reserves. Both the main fault zone and secondary faults along the strike-slip fault exhibit enhanced oil and gas displays, demonstrating robust fault-controlled storage and enrichment characteristics [1,2]. Nevertheless, the exploration of ultradeep fault-controlled carbonate reservoirs is exceedingly complex, particularly within strike-slip fault fracture zones, where reservoirs exhibit significant heterogeneity [3,4]. The reservoir characteristics suggest that the storage spaces are predominantly influenced by dissolution or tectonic factors [5]. Therefore, the identification of strike-slip fault zones has emerged as a critical challenge in seismic exploration. The burial depth of Ordovician oil and gas reservoirs in western China

**Data availability statement:** The data relevant to this work, including the seismic and logging data, are owned by the Yueman Oilfield of the Tarim Branch of China National Petroleum Corporation and authors do not have permission to share the data publicly. To apply for access, please contact the corresponding author of this paper via email or Ma Zhisong, a Senior Engineer at PetroChina Tarim Oilfield Company, at mazhisong5426@gmail.com.

**Funding:** The author(s) received no specific funding for this work.

**Competing interests:** The authors have declared that no competing interests exist.

ranges from 7500 to 9000 m, placing them in the ultradeep category. In addition, the quality of the seismic data for the target layer section was suboptimal. In the seismic profiles, the fracture and broken zones of the Ordovician carbonate reservoirs exhibit varying energy and chaotic reflection characteristics. These features are strip-shaped on the plane [6,7]. In the exploration of Ordovician carbonate fractured reservoirs in western China for oil and gas, seismic reflections characterized by intense energy 'beads' are predominantly aligned with high-yield oil and gas wells [8], whereas strip-shaped chaotic seismic reflections are indicative of high-quality reservoir characteristics. Existing technologies excel in accurately identifying strong 'bead' reflections on seismic profiles, including techniques such as seismic attribute, coherence, and curvature analyses. However, the direct identification of strip-shaped chaotic reflections is challenging. Currently, various seismic attributes are employed in fault detection, including measures such as the similarity [9] and coherence [10,11] of seismic reflection continuity, as well as the analysis of variance [12], curvature [13,14], and gradient amplitude [15] associated with reflection discontinuity.

These attributes exhibit certain efficacy in identifying primary strike-slip faults but are less effective for secondary faults. Despite leveraging the seismic response characteristics of strike-slip faults, employing methods such as root mean square amplitude, amplitude gradient, multi-data fusion prediction [16], and structural tensor [17] for strike-slip fault identification remains limited for effectively distinguishing strong bead-shaped reflections, with less clarity in identifying strip-shaped chaotic reflection characteristics. To enhance fault features and distinguish faults from other seismic image features, a commonly used approach involves the use of a vertically stretched window during seismic coherence calculations. This method assumes that faults are more vertically oriented than other geological features, thereby suppressing the latter. However, faults are rarely strictly vertical and may not necessarily be perpendicular to the seismic reflection surface. Pedersen et al. [18,19] proposed an ant-tracking method that enhanced fault features along 'artificial ant' paths, assuming that these paths corresponded to faults. Alternative approaches suggested by other scholars [20,21] involve smoothing along fault trends and dips to enhance the fault features by exploring various combinations of fault trends and dips. Hale [22] calculated the fault possibility or orientation similarity by smoothing both the numerator and denominator of the similarity along fault trends and dips. Nevertheless, these methods often yield multiple solutions and are susceptible to illusions that do not align with the structural shadows. Therefore, this study examined the fusion of multiple preferred attributes for attribute fusion data volumes using the optimal surface voting algorithm [23] to solve the global maximization optimal paths and surfaces for multi-attribute fusion data volumes. Compared to traditional methods, this method fuses multiple preferred attributes to overcome the limitations of single attributes and improve the accuracy of strike-slip fault interpretation. Furthermore, the optimal surface voting algorithm can calculate smooth paths or surfaces from high-noise or discontinuous features to improve the fracture characterization accuracy and continuity. The application shows that this method can effectively identify strong energy bead-shaped reflections and chaotic seismic reflection characteristics with clear effects on identifying strike-slip faults.

## Methodology

The Yueman Block is situated northwest of the Shuntuo Low Rise, connected to the north by the Halahatang Depression. Its southern boundary of the northern depression is bordered by Manjia'er and Awati and is surrounded by hydrocarbon-generating depressions (Fig 1). The area boasts multiple oil-bearing strata, with Ordovician carbonate reservoirs particularly noteworthy for their extensive and localized oil accumulation. The reservoirs in this area are primarily fracture-controlled carbonate reservoirs and their distribution is influenced

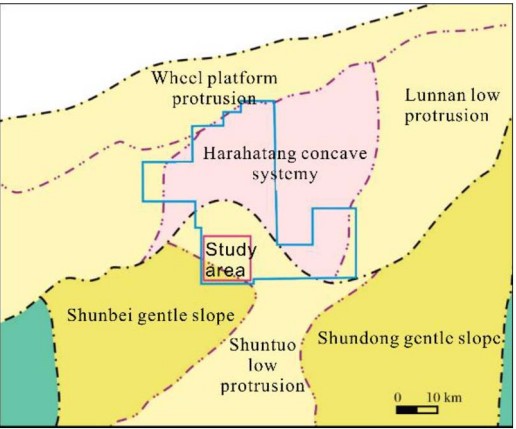

**Fig 1.  Block structure location in the Yueman area.**

by strike-slip faults. The seismic data quality of the target layer segment in the study area was poor, with fault breakage zones exhibiting uneven energy and chaotic reflection characteristics in the seismic profiles. Several strike-slip faults were indistinctly identified, and the fault distance was not apparent. The low-velocity zones caused by faults resulted in bead-like reflections.

This study integrated the developmental characteristics of faults in the research area with prior geological insights to establish models of strike-slip faults at different scales. Forward simulations were conducted to serve as a foundation for attribute optimization in strike-slip fault identification [24–26]. Diverse seismic attributes were calculated and optimized based on the forward data volume. Multiple seismic attributes with the most effective identification outcomes were selected, and the attribute ratio fusion method was employed for a comprehensive evaluation of the strike-slip faults.

Subsequently, utilizing the attribute fusion data volume, a dynamic path optimization algorithm was employed for precise fault characterization. The results of the path optimization calculations were combined with the attribute fusion data volume to compute the fault probability attribute. This not only enhanced the accuracy of strike-slip fault identification, but also improved the continuity of fault characterization. Finally, the accuracy of the proposed method was verified by comparing the fault distribution of the model with the fault identification results. This method was applied to actual seismic data to demonstrate its practical utility.

## Attribute optimization based on the forward model

**Establishment of forward model.**  The pattern of strike-slip faults is complex, and seismic anomalies manifest in diverse forms. To assess the efficacy of this method in a three-dimensional data volume, this study employed a three-dimensional strike-slip fault geological model (Fig 2a) for forward simulation. Simulated seismic records were obtained through wave equation forward modeling using a real velocity model. After reverse time migration processing, post-stack Laplace filtering, and FK domain filtering, post-stack seismic data (Fig 2b) were generated. The seismic data had a sampling interval of 6 m and trace spacing of 12.5 m. The seismic source wavelet is a key factor that influences the forward simulation results. In this study, we conducted a forward simulation of a fracture-cavity model under real conditions. We extracted statistical wavelets based on the original seismic data from the target layer, with a sampling interval of 2 ms and a dominant frequency of 25 Hz. Building upon this

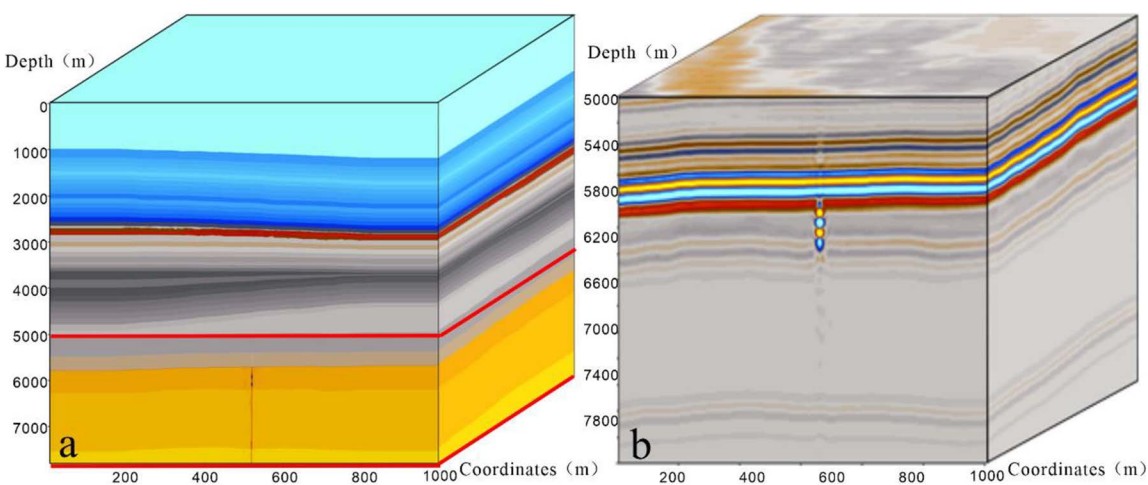

**Fig 2. Comparison between forward model data volume and forward geological model.** (a) Geological model; (b) Reverse time migration data volume from the forward model.

foundation, fracture identification was performed using multiple attributes. Attributes were optimized based on their identification effectiveness, providing a foundation for subsequent research.

**Fracture attribute analysis.** There are various types of seismic data, each with distinct advantages and limitations in detecting seismic faults. Seismic anomalies within strike-slip fault zones are diverse and are difficult to characterize accurately using a single attribute. The adoption of the multi-attribute fusion method proved effective in enhancing the accuracy and reliability of seismic fault identification. This study employed a multi-attribute fusion approach to comprehensively analyze different types of data, aiming for a more precise determination of key information, such as the location, shape, and nature of seismic faults. Different seismic attributes were screened in the attribute fusion process. By optimizing the attributes, the most representative and predictive attributes were selected, and irrelevant or redundant attributes were eliminated to reduce complexity. Attribute optimization serves a dual purpose: it reduces computational costs while enhancing the accuracy and reliability of attribute fusion data, thereby improving prediction accuracy. This underscores the crucial role of attribute optimization as a key component influencing the overall performance and effectiveness of the model. Commonly employed attributes in fracture identification include the coherence, curvature, amplitude, and amplitude gradient tensors. This study investigated the seismic response characteristics of different reservoir configurations based on typical fracture-controlled carbonate rock models and forward data. Distinct seismic attributes and characterization techniques have been applied to various unit configurations.

(1) Coherence Attribute: Currently, three algorithms are applied for coherence: first-generation coherence has the fastest calculation speed, but its noise resistance is the worst; it is not suitable for low signal-to-noise ratio data in deep layers in research areas; the second-generation coherence algorithm is slightly slower than the first-generation coherence algorithm, but its noise resistance has improved; it is more suitable for large work areas with low signal-to-noise ratios, and the third-generation or eigenvalue coherence algorithm can eliminate factors such as multipath and lateral scattering that affect coherence. It improves resolution and clarity but is sensitive to seismic noise; therefore, it requires high quality data.

(2) Curvature Attribute: Curvature is a two-dimensional attribute of a curve; it is the inverse radius of a circle, and can reflect how curved an arc is, with a larger curvature indicating more curvature. For brittle rocks, the fracture development is proportional to the curvature; therefore, the structural curvature can be used to predict the fracture development. The structural curvature in three-dimensional seismic interpretation is calculated based on the layer position; it reflects how curved any point is on an interpreted layer position.

(3) Gradient Structure Tensor Attribute: A karst body is a combination of caves, karst holes, and fissures. Conventional fracture detection attributes, such as fine coherence or ant bodies, mainly characterize internal fracture distribution features within karst bodies; they cannot accurately characterize the outline features within karst body closures. The gradient structure tensor is a mathematical tool used to describe local structural features in images or signals. When processing seismic images, local structural features can be extracted by calculating the gradient vectors and structure tensors. By calculating the principal axis direction and size from the gradient tensors, we can determine the potential fault locations and directions in images, thereby assisting in identifying and locating seismic faults. This method can improve the accuracy and efficiency of seismic image processing, and has important application value for seismic exploration.

(4) Amplitude Attribute: The amplitude attribute is one of the most basic attributes of seismic waveforms; it refers to the amplitude of the seismic waveforms. Seismic faults typically induce phenomena such as reflection, refraction, or diffraction when waves propagate underground, resulting in varying amplitudes. By analyzing the changes in the waveform amplitude, we can extract essential information such as the location or shape of seismic faults. Processing the amplitude attributes enhances the accuracy and reliability of seismic fault identification. In our model, we utilized the coherence, curvature, amplitude, and gradient structure tensor attributes for fracture identification (Fig 3).

The presence of a low-speed zone caused by fractures leads to bead-like reflections, resulting in errors in traditional coherence and curvature attribute identification of strike-slip fractures. Fig 2a represents the geological model, whereas Fig 3b depicts the seismic forward modeling data volume. A comparison between the coherence attribute (Fig 3c) and forward model profiles (Fig 3a) reveals poor coherence on both sides of the bead-like reflections, indicating a mismatch with the fracture location. Similarly, comparing the curvature attribute profile (Fig 3d) with the forward model profile (Fig 3a) shows an increase in curvature on both sides, causing a significant error in identifying the location of the strike-slip fractures.

The gradient structure tensor attribute (Fig 3e), when compared with the forward model profile (Fig 3a), accurately characterized the outline of the karst body, highlighting high-value areas consistent with the reflection anomalies. This observation reflects the comprehensive influence of various elements on karst body configuration. Significant bead-like reflections within the strike-slip fractures were evident in the amplitude attributes (Fig 3f). Although the amplitude attribute can provide information on the location or shape of strike-slip fractures, relying solely on this attribute has limitations in accurately identifying strike-slip faults. Therefore, this study employed a multi-attribute fusion method that significantly improved the accuracy of strike-slip fracture identification.

**Attribute fusion.** The commonly used methods for attribute fusion include pattern recognition, ratio fusion, and kernel principal component analysis. Ratio fusion adds weights to different data attributes based on certain proportions to obtain the final results. Comparatively, the ratio fusion operation is convenient and fast, and the effect is excellent. Based on a previous analysis, this study optimized the gradient structure tensor amplitude attributes for attribute fusion. Because the value range difference between the structure

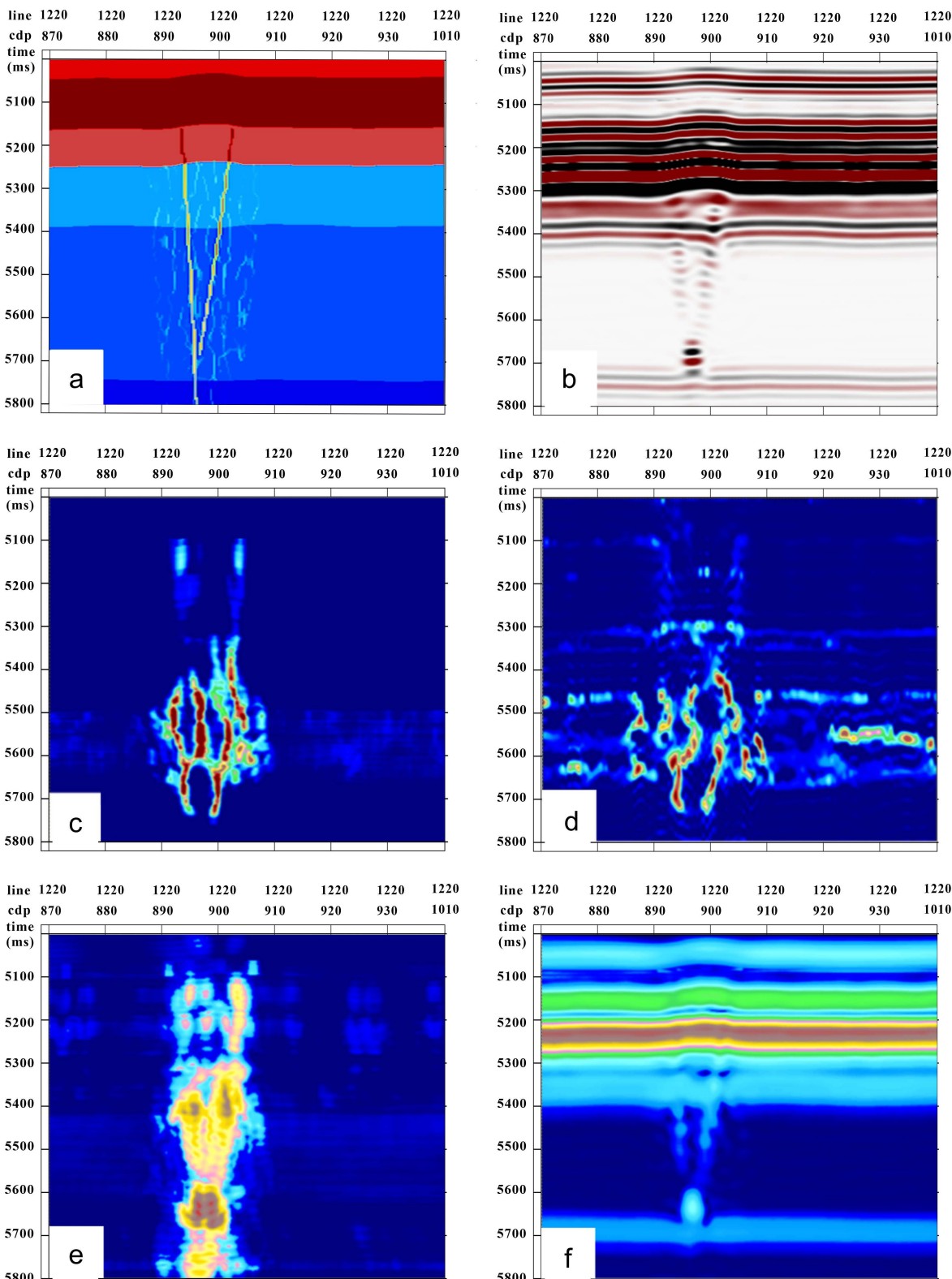

**Fig 3. Attribute fracture recognition profiles compared to the forward model profile.** (a) Geological model; (b) seismic forward modeling data volume, with a sampling rate of 2 ms and dominant frequency of 25 Hz; (c) coherence attribute profile; (d) curvature attribute profile; (e) gradient structure tensor attribute data volume; and (f) amplitude attribute.

gradient tensor amplitude attributes is extremely large, direct fusion covers the structure gradient tensor feature, making it impossible to accurately identify local structural features within strike-slip fractures from the fused property data. Therefore, we first normalized the structural gradient tensor amplitude properties by scaling both property value ranges into the same interval. This ensures that complete feature retention is possible for strike-slip fractures during the data fusion process.

Finally, after the normalization treatment of the gradient structure tensor amplitude properties, the two earthquake property volumes were combined using a 1:1 proportional relationship ratio, generating a new fused property volume relative to a single property volume. This volume could more accurately identify strike-slip fractures.

## Fracture probability calculation based on fused attributes

In fused-property images, faults are typically identified using relatively low or high values. Therefore, picking faults from property images is viewed as a challenge in determining the optimal path through the global minimum or maximum values. Traditionally, earthquake property images can detect most fault locations; however, they also highlight noise. In addition, traditional earthquake property images exhibit poor fault feature continuity. Therefore, this study innovatively used a dynamic programming algorithm to efficiently extract the optimal path within a window range by calculating the fracture probability based on the path extraction repetition rate property value. Eventually, a strike-slip fracture probability image was formed. In addition to achieving high-precision identification of strike-slip fractures, the continuity within the fracture image was improved.

To extract continuous fault information from fault properties, this study first transposed the image into a new space, where the vertical axis represents the earthquake trace and the horizontal axis represents the depth. In addition, it discretely selected a certain number of seed points within the fault property image to indicate the position at which the fault surface passes. It set a zero value (minimum value) within a conical region above and below the seed points, and set one value (maximum value) near the seed points to better select the optimal fault passing through the global maximum fault property value.

Because transposing the image does not actually improve the fault features within the image, before selecting fault surfaces, nonlinear smoothing technology was applied to smooth the property image. This involves obtaining a smoothed image using low-slope constraint nonlinear smoothing, ensuring smooth continuity of earthquake property features while weakening noise, and making fault features more evident and continuous.

To solve the constrained maximization problem and determine the optimal path, this study used a nonlinear smoothing forward integration backward tracking dynamic programming algorithm [27]. The first step in smoothing enhances features related to selected faults or paths while weakening noise features unrelated to faults. The smoothing filter is achieved by applying nonlinear integration to the forward and backward input fault property images.

The forward accumulation (from left to right) of input image $g[i,j]$ with slope constraint $|j[i+1]\text{-}j[i]|$ can be implemented in the following way:

$$f[0,j] = g[0,j] \tag{1}$$

$$f[i,j] = g[i,j] + max \begin{cases} f[i-d,j-1] + \sum_{k=i-d+1}^{i-1} g[k,j-i] \\ f[i-1,j] \\ f[i-d,j+1] + \sum_{k=i-d+1}^{i-1} g[k,j+1] \end{cases} \tag{2}$$

where $i = 1,2,\ldots,N-1$; $d$ is the integer closest to $1/\varepsilon$; $g[i,j]$ is an input fault attribute image; and $f[i,j]$ is the forward accumulated image. This forward accumulation can be considered as a one-sided smoothing filter. As the max function is used in the above equation, it is nonlinear. Similarly, the backward accumulation (from right to left) of input image $g[i,j]$ with a slope constraint can be implemented as follows:

$$b[N-1,j] = g[N-1,j] \tag{3}$$

$$b[i,j] = g[i,j] + max \begin{cases} b[i+d,j-1] + \sum_{k=i+1}^{i-1} g[k,j-i] \\ d[i+1,j] \\ b[i+d,j+1] + \sum_{k=i+1}^{i-1} g[k,j+1] \end{cases} \tag{4}$$

where $d = [1/\varepsilon]$, which is the integer closest to $1/\varepsilon$; and $b[i,j]$ is the backward accumulated image. Bilateral nonlinear smoothing is defined as a combination of forward and backward accumulation.

$$s[i,j] = f[i,j] + b[i,j] - g[i,j] \tag{5}$$

In this process, $g[i,j]$ must be subtracted, because it is calculated for both $f[i,j]$ and $b[i,j]$. Using forward and backward accumulation with slope constraints, low-slope constraints can be imposed in nonlinear smoothing to produce smooth and continuous features in the images. After smoothing, noise features are weakened, whereas image features related to faults are more evident and continuous in the middle of the output image, making it easier to select faults or maximum paths. In the smoothing process, the areas above and below the control points are set to zero, whereas the values near the control points are set to one to ensure that the next step of maximum path picking finds a path through the control points. Through this process, the local optimal paths passing through each seed point can be obtained. Most of these selected paths correctly follow fault attribute features, and certain paths are traversed multiple times, indicating that they may follow real faults. Paths traversed a few times or only once indicate that the path is a false image caused by noise rather than a real fault. The higher the overlap of the paths, the higher the probability of fracture. In this study, the attribute values and path overlap were comprehensively integrated to calculate the fracture probability using the following formula:

$$k^*[i,j] = \frac{k[i,j] - k_{min}}{k_{max} - k_{min}} \tag{6}$$

$$g^*[i,j] = \frac{g[i,j] - g_{min}}{g_{max} - g_{min}} \tag{7}$$

$$P = \begin{bmatrix} k_{11}^* & \cdots & k_{m1}^* \\ \vdots & \ddots & \vdots \\ k_{1n}^* & \cdots & k_{mn}^* \end{bmatrix} \odot \begin{bmatrix} g_{11}^* & \cdots & g_{m1}^* \\ \vdots & \ddots & \vdots \\ g_{1n}^* & \cdots & g_{mn}^* \end{bmatrix} \tag{8}$$

where $k[i,j]$ represents the number of path traversals, $g[i,j]$ represents the input attribute value, and $P$ represents the calculated fracture probability.

## Fracture identification based on forward modeling data

Based on the fusion data volume of the amplitude attribute data and gradient structure tensor attributes, we utilized an optimal surface voting algorithm for strike-slip fracture identification. By comparing the results of the strike-slip fracture identification with the forward models (Fig 4), the optimal surface voting algorithm is shown to accurately characterize the fracture locations. Moreover, it effectively distinguishes between two closely positioned fractures. Simultaneously, this method is capable of suppressing noise and providing a clearer and more continuous characterization of fracture distribution.

On this basis, this study performed thinning processing on seismic records to obtain a three-dimensional seismic data volume with a trace spacing of 25 m and used optimal surface voting fracture identification technology for fracture detection. As shown in Fig 5, as the trace spacing increased, the fracture detection results were significantly affected.

The impact of trace spacing on the fracture detection results is evident, with a smaller trace spacing leading to a higher recognition accuracy. When seismic records with a trace spacing of 25 m were analyzed, fracture detection identified two closely positioned fractures as a single fracture, and small fractures could not be accurately identified. However, for a trace spacing of 12.5 m, the recognition accuracy significantly improved. This study conducted fracture detection along the Ying Mountain Group (TO12y) and Peng Luan Group (TO1p) by extracting equal-depth slices from the optimal volume of surface voting fracture identification data. These results were then compared with the forward model stratum slices (Fig 6), revealing a substantial alignment between the fracture detection results and the forward models.

## Real data processing

The Tarim Basin, a vast composite basin comprising Paleozoic craton basins and Mesozoic-Cenozoic foreland basins, is characterized by extensive Ordovician carbonate reservoirs, making it the largest inland oil and gas basin in China. During its early exploration and development, emphasis was primarily placed on ancient uplifts. However, with significant advancements in the understanding of deep to ultradeep Ordovician carbonate reservoirs, attention has shifted to fault-controlled dissolution body oil and gas reservoirs, marking a focal point in oilfield exploration and development [28–29]. The focus of reservoir understanding has gradually transitioned to fault-controlled storage and control, in which the identification of strike-slip faults is crucial

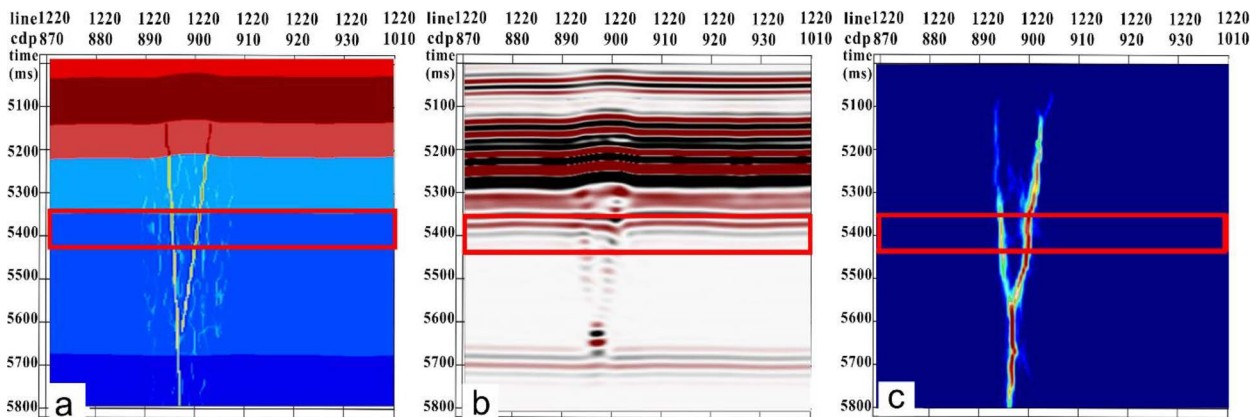

**Fig 4. Comparison of optimal surface voting fracture identification profile with forward model profile.** (a) Forward model profile; (b) Seismic record profile; and (C) Optimal surface voting fracture identification profile.

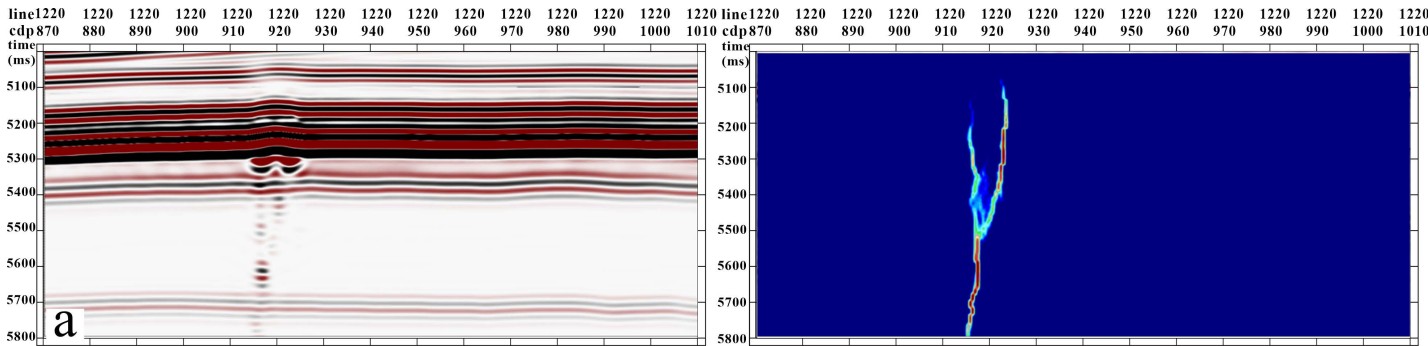

**Fig 5. Comparison of seismic record with optimal surface voting fracture identification profile with 25 m trace spacing.** (a) Seismic record profile with 25 m trace spacing; (b) Optimal surface voting fracture identification profile with 25 m trace spacing.

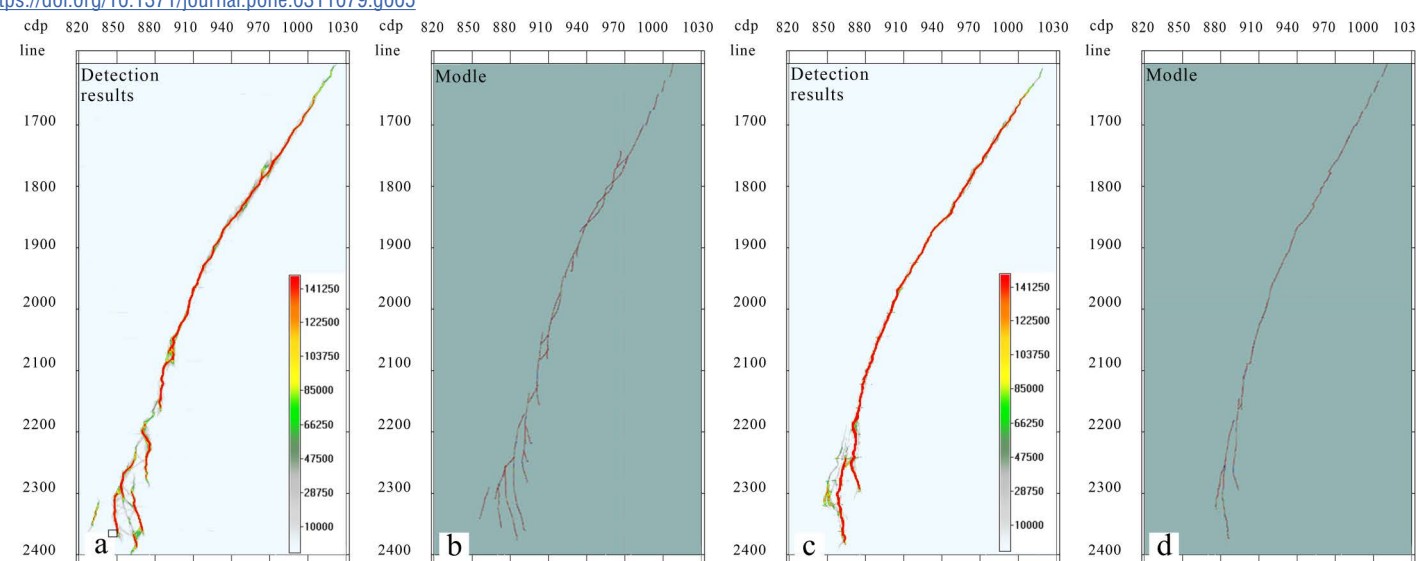

**Fig 6. Comparison of equal depth slices from optimal surface voting fracture identification with forward model slices.** (a) equi-depth slice of the model at 7000 m; (b) fault detection results of our method at 7000 m; (c) equi-depth slice of the model at 7800 m; and (d)fault detection results of our method at 7800 m.

for guiding oil and gas resource exploration and development. This study used the Yueman area as an example to assess the application effectiveness of the proposed multi-attribute optimal surface voting fault identification technology. The forward model data profiles exhibited characteristics similar to the actual seismic data and effectively captured the diverse seismic anomaly characteristics of the strike-slip fault zones within the study area (Fig 7).

The quality of the target layer segment seismic data in the study area was poor. Fault breakage zones show uneven energy and chaotic reflection characteristics in their seismic profiles. Several strike-slip faults were not clearly identified, and the fault distance was not clear. Low-velocity zones caused by faults result in bead-like reflections (Fig 8).

To address the diverse seismic anomalies in the Yueman area strike-slip fault zone, where a single attribute falls short of fully characterizing the complete features of the strike-slip fault, we integrated the test results from the forward models. Specifically, the amplitude attributes were selected because they demonstrated superior identification of bead-like reflection amplitude anomalies. This selection facilitated the determination of the fault location and morphology. Concurrently, high values from the gradient structure tensor were found to align with the

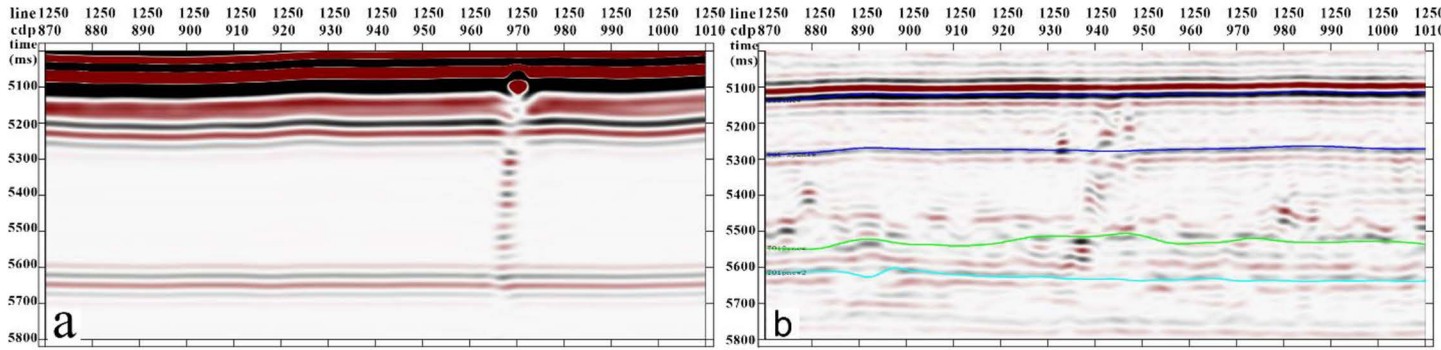

**Fig 7. (a) the forward-simulated seismic data and (b) actual seismic data from the study area.** Both exhibit certain characteristic similarities. However, it is evident from the actual data that the seismic data quality of the target layer segment in the study area is poor.

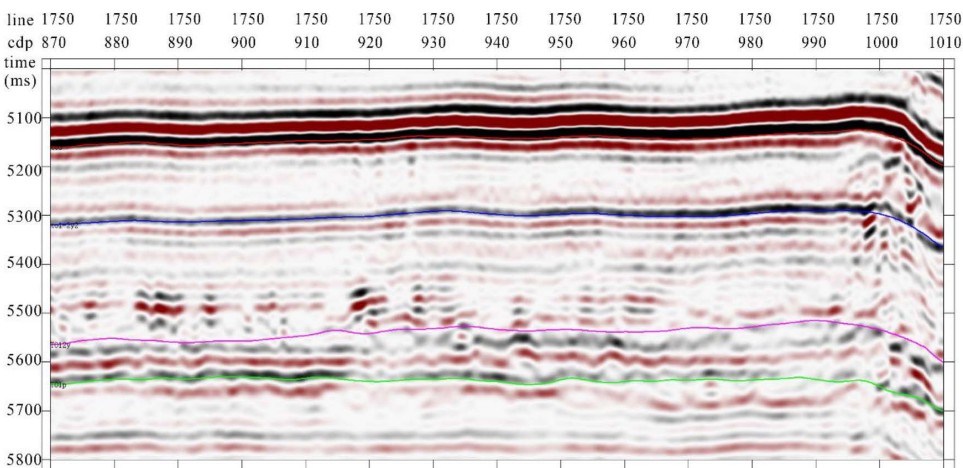

**Fig 8. Tarim Basin area seismic record (profile).** The energy of the fault breakage zones is uneven, and the reflection characteristics are chaotic.

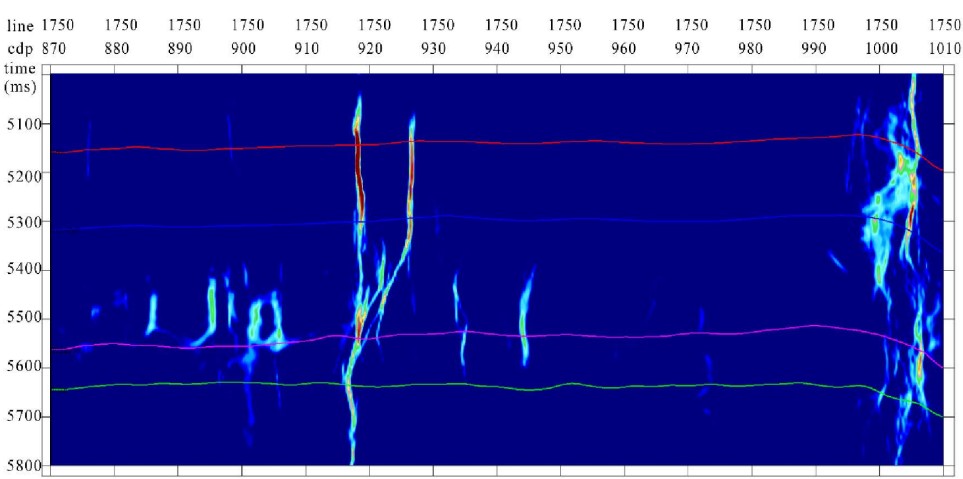

**Fig 9. Tarim Basin area multi-attribute optimal surface voting fault identification data volume (profile).**

reflection anomaly areas of the dissolution bodies, indicating the comprehensive influence of various elements within the dissolution body configuration. To enhance strike-slip fault identification, we employed an optimal surface voting algorithm based on the fusion of attribute data volumes (Fig 9).

Overlaying the Tarim Basin Yueman Region strike-slip fault identification data volume with the seismic record data (Fig 10), this technology can effectively identify subtle waveform changes caused by strike-slip faults while suppressing noise achieving accurate characterization of strike-slip faults.

Multi-attribute optimal surface voting fault identification slices were extracted along Ying Mountain Group (TO12y) and Peng Lai Group (TO1p) (Fig 11). Noise in the original data was suppressed, whereas the continuity was strong in the fault-identification results with high accuracy.

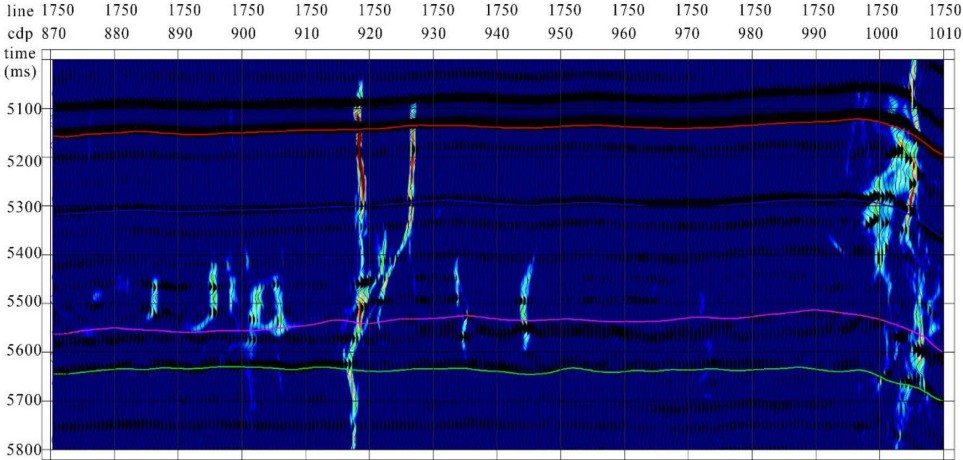

**Fig 10. Tarim Basin area strike-slip fault identification data volume overlaid with seismic record (profile).**

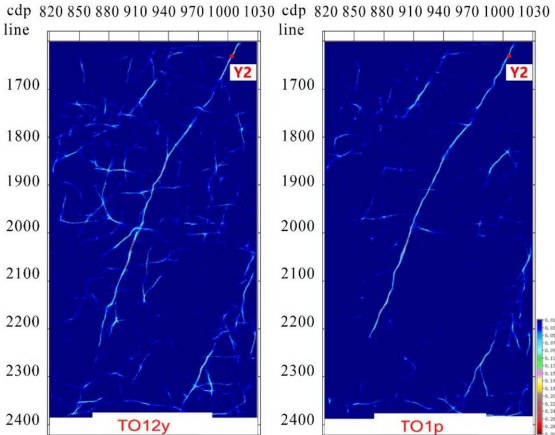

**Fig 11. Tarim Basin Yueshan area strike-slip fault identification along layer slice.** The fault identification results are clear and can effectively identify minor faults.

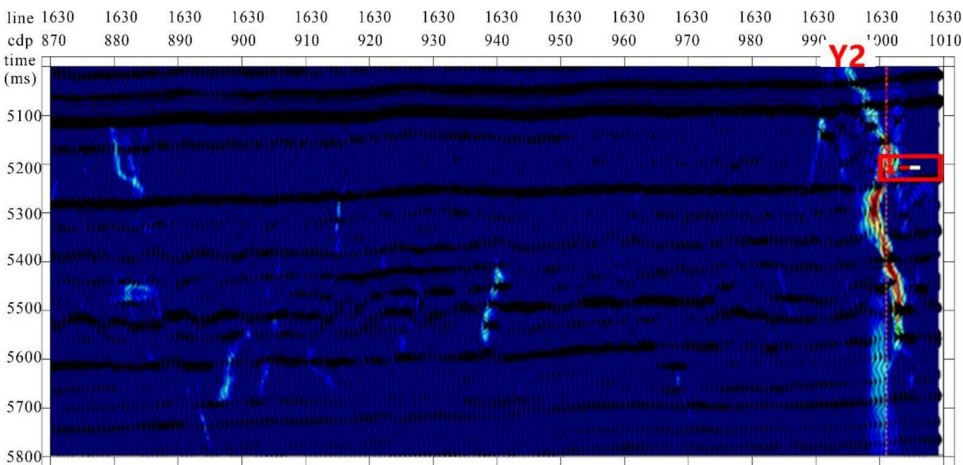

**Fig 12. Strike-slip fault identification profile passing through well Y2 within study area.** The fault identification results are consistent with the seismic profile characteristics.

Based on the strike-slip fault identification, the profile passing through well Y2 was extracted within the study area (Fig 12). Combined with logging interpretation data and actual production data, Y2 encountered a strike-slip fault zone, obtaining an oil layer thickness of 9.5 m with a cumulative production exceeding 100000 tons. Therefore, the method combining fusion attributes with the dynamic optimization optimal surface recognition algorithm can achieve accurate recognition of strike-slip faults, providing a technical guarantee for exploration and development in the Yueman area.

## Discussion

Strike-slip faults typically exhibit minimal observable changes in distance on seismic profiles, and their structures do not cause interference or amplitude alterations in the reflected waves. This makes the identification of strike-slip faults challenging using traditional seismic processing methods, such as coherence, variance, and curvature. To address the diverse seismic anomalies within strike-slip fault zones where a single attribute falls short of fully characterizing the complete features of strike-slip faults, this study focused on selecting amplitude attributes based on forward-simulated records. These attributes proved to be more effective in identifying bead-like reflection amplitude anomalies, aiding in the determination of their location and morphology. Additionally, the high values of the gradient structure tensor were consistently aligned with the reflection anomaly areas of the dissolution bodies, reflecting the comprehensive influence of various elements within the dissolution body configuration.

In practical applications within the exploration and development of the Yueman area in the Tarim Basin, particularly in controlled-type carbonate rock reservoirs, the method successfully identified subtle waveform changes caused by strike-slip faults while suppressing noise. This approach achieved an accurate characterization of strike-slip faults, providing a technical guarantee for the exploration and development of the Yueman area.

Despite the significant advancements presented in this study, several challenges and areas for further research remain. One key challenge is the variability in seismic data quality, which can impact fault detection accuracy. High-noise environments and poor data quality can obscure fault signatures, complicating the effective application of the proposed multi-attribute fusion method. Future research should focus on enhancing noise suppression techniques

and improving data acquisition methods to ensure higher fidelity in seismic data. Integrating this method with advanced geophysical techniques, such as machine learning and artificial intelligence, could significantly enhance fault detection capabilities. Machine learning algorithms, for instance, could be trained on large datasets to recognize complex fault patterns and improve the accuracy of attribute selection and fusion processes. Interdisciplinary approaches like these could lead to the development of more sophisticated and automated fault detection systems, ultimately improving exploration and production efficiency.

In conclusion, the dynamic optimization-based multi-attribute fusion method represents a substantial advancement in accurately identifying strike-slip faults in complex geological settings. By addressing the challenges of seismic data variability, exploring scalability across different geological contexts, integrating advanced geophysical techniques, and considering economic factors, future research can further refine and expand the applicability of this promising approach.

## Author contributions

**Data curation:** Chen Ma, Youcai Tang, Suo Cheng.

**Methodology:** Youcai Tang.

**Project administration:** Chao Wang.

**Validation:** Xin Wang.

**Writing – original draft:** Chen Ma.

**Writing – review & editing:** Handong Huang.

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
