## [Decision Letter · Decision Letter 0]

24 Mar 2024

PONE-D-24-05733Research and Application of a Dynamic Optimization-Based Multi-Attribute Fusion Method for Fault DetectionPLOS ONE

Dear Dr. Huang,

Thank you for submitting your manuscript to PLOS ONE. After careful consideration, we feel that it has merit but does not fully meet PLOS ONE’s publication criteria as it currently stands. Therefore, we invite you to submit a revised version of the manuscript that addresses the points raised during the review process. Please answer all the reviewer's concern and follow all the suggestions in order to improve the manuscript.

We look forward to receiving your revised manuscript.

Kind regards,

Fabio Trippetta, Ph.D.

Academic Editor

PLOS ONE

4. Please update your submission to use the PLOS LaTeX template. The template and more information on our requirements for LaTeX submissions can be found at http://journals.plos.org/plosone/s/latex.

5. For studies involving third-party data, we encourage authors to share any data specific to their analyses that they can legally distribute. PLOS recognizes, however, that authors may be using third-party data they do not have the rights to share. When third-party data cannot be publicly shared, authors must provide all information necessary for interested researchers to apply to gain access to the data. (https://journals.plos.org/plosone/s/data-availability#loc-acceptable-data-access-restrictions)

4) All necessary contact information others would need to apply to gain access to the data.

6. PLOS requires an ORCID iD for the corresponding author in Editorial Manager on papers submitted after December 6th, 2016. Please ensure that you have an ORCID iD and that it is validated in Editorial Manager. To do this, go to ‘Update my Information’ (in the upper left-hand corner of the main menu), and click on the Fetch/Validate link next to the ORCID field. This will take you to the ORCID site and allow you to create a new iD or authenticate a pre-existing iD in Editorial Manager. Please see the following video for instructions on linking an ORCID iD to your Editorial Manager account: https://www.youtube.com/watch?v=_xcclfuvtxQ.

Reviewers' comments:

Reviewer's Responses to Questions

**Comments to the Author**

1. Is the manuscript technically sound, and do the data support the conclusions?

Reviewer #1: Partly

Reviewer #2: Yes

2. Has the statistical analysis been performed appropriately and rigorously? 

Reviewer #1: N/A

Reviewer #2: Yes

3. Have the authors made all data underlying the findings in their manuscript fully available?

Reviewer #1: No

Reviewer #2: Yes

4. Is the manuscript presented in an intelligible fashion and written in standard English?

Reviewer #1: No

Reviewer #2: Yes

5. Review Comments to the Author

Reviewer #1: The scientific research article is highly intriguing, also perhaps because of my scientific background; however, some major modifications are needed to refine the paper presentation and clarity of the concepts discussed. Authors will find the following comments also in the attached pdf file to be facilitated in revisions.

1-I recommend the authors to modify the title of the paper in “Application of a Dynamic Optimization-Based Multi-Attribute Fusion Method for Fault Detection”, which is clearer and more concise.

2-In the abstract is mentioned the Yueman area. Since the Tarim Basin covers an area of 888,000 km2, and since there is no geographic (with an attached location map) and geological framework in the paper, it would be explanatory to at least place indications of the cardinal points of this area.

3-row 6: rephrase “[…] strike-slip fault zones in ultra-deep dense limestone is a pressing issue […]” in “strike-slip fault zones in ultra-deep tight limestone is a critical issue”.

4-row 8: I do not understand what the authors mean by 'beaded'. I suggest finding a clearer synonym.

5-row 12: perhaps the term 'merged' is more appropriate than 'fused'.

6-rows 16-18: this sentence is not useful as the location has already been declared as well as the visualisation is improved. I advise the authors to delete it.

7-About the Keywords: I advise the authors to work better on keywords. Since search engines often identify works by their title, abstract and keywords, authors will be more likely to be read if they use the right and non-repeated keywords among the search items. E.g., "strike-slip fault" is already present three times in the abstract, maybe another keyword could be more suitable.

8- At the very beginning of the Introduction paragraph, where authors state “[…] The deep-seated strike-slip faults in western China play a pivotal role in controlling substantial reserves of oil and gas. Both the main fault zone and secondary faults along the strike-slip fault exhibit enhanced displays of oil and gas, demonstrating robust fault-controlled storage and enrichment characteristics. […]”, Citations are needed to support theese sentences. Recommendation:

- Zhao, R., Zhao, T., Kong, Q., Deng, S. and Li, H., 2020. Relationship between fractures, stress, strike-slip fault and reservoir productivity, China Shunbei oil field, Tarim Basin. Carbonates and Evaporites, 35, pp.1-14.

- Deng, S., Zhao, R., Kong, Q., Li, Y. and Li, B., 2022. Two distinct strike-slip fault networks in the Shunbei area and its surroundings, Tarim Basin: Hydrocarbon accumulation, distribution, and controlling factors. AAPG Bulletin, 106(1), pp.77-102.

9- Continuing in the introductions, where authors state “[…] Nevertheless, exploring ultra-deep fault-controlled carbonate reservoirs proves to be exceedingly complex, particularly within the strike-slip fault fracture zone where reservoirs exhibit significant heterogeneity. Reservoir characteristics suggest that storage spaces are predominantly influenced by dissolution or tectonic factors. […]”, Also this sentence needs citations, authors should cite:

- Tomassi, A., Trippetta, F., de Franco, R. and Ruggieri, R., 2023. How petrophysical properties influence the seismic signature of carbonate fault damage zone: Insights from forward-seismic modelling. Journal of Structural Geology, 167, p.104802.

- Tomassi, A., Milli, S. and Tentori, D., 2024. Synthetic seismic forward modeling of a high-frequency depositional sequence: The example of the Tiber depositional sequence (Central Italy). Marine and Petroleum Geology, 160, p.106624.

10- in rows 27-29, when authors state “[…] Reservoir characteristics suggest that storage spaces are predominantly influenced by dissolution or tectonic factors. […]”, usually, in scientific publications, introductions and discussions are the most citation-rich paragraphs. Especially in introductions in which the research question is contextualised in the international scenario, almost every sentence written by the authors has to be justified by the literature unless it is a 'discovery' of the authors themselves illustrated for the first time in the paper in question. It seems strange to me how there is this enormous lack of citations here. I give a tip on citation:

- Radwan, A.E., Trippetta, F., Kassem, A.A. and Kania, M., 2021. Multi-scale characterization of unconventional tight carbonate reservoir: Insights from October oil filed, Gulf of Suez rift basin, Egypt. Journal of Petroleum Science and Engineering, 197, p.107968.

11- row 37: I do not understand what the authors mean by energy 'beads'. Perhaps that these reflectors have marked amplitudes caused by the strong acoustic impedance contrast?

12- in rows 37-38, The authors should explain why these reflectors denote high-quality reservoir (even referring to literature).

13- in row 39 authors state “[…] existing technologies excel in accurately identifying strong ’beads’ reflections […]”, in my opinion authors should mention which technologies.

14- The second paragraph should be a geological setting section. The authors need to add a paragraph on the geographical-geological-stratigraphic-geodynamic framework of the area under examination, which is totally lacking. A case study such as the one examined in this paper certainly deserves to be valorised in order to also contextualise the research and be applicable (and reproducible) to other similar realms worldwide, also giving a broader scope to the work.

15- in rows 76-77, authors state “[…] Forward simulations are then conducted to serve as a foundation for attribute optimization in strike-slip fault identification. […]”. When stating concepts already well established in the literature, authors are requested to add references. Authors should cite here:

- Alaei, B., 2012. Seismic modeling of complex geological structures. Seismic Waves-Research and Analysis, 11, pp.528-529.

- Tomassi, A., Trippetta, F., de Franco, R. and Ruggieri, R., 2022. From petrophysical properties to forward-seismic modeling of facies heterogeneity in the carbonate realm (Majella Massif, central Italy). Journal of Petroleum Science and Engineering, 211, p.110242.

- De Franco, R., Petracchini, L., Scrocca, D., Caielli, G., Montegrossi, G., Santilano, A. and Manzella, A., 2019. Synthetic seismic reflection modelling in a supercritical geothermal system: An image of the k-horizon in the larderello field (Italy). Geofluids, 2019.

16- in paragraph “2.1.1. Establishment of Forward Model”, It would be very interesting if the authors also stated the central frequency of acquisition and the vertical and horizontal resolution, which are decisive factors in fault detection especially when dealing with very deep sub-vertical faults. Authors are requested to add this information.

17- Figure1: This caption is very unexplanatory. The authors should also briefly describe what is seen in the volumes represented. They should also state in the image which is Figure 1A and which is Figure 1B.

18- in rows 117 to 119, authors state: “[…] The paper investigates the seismic response characteristics in different reservoir configurations based on typical fracture-controlled carbonate rock models and forward data. […]”. I understood this to be the case study of the Tarim Basin. Perhaps the authors meant that this workflow is also applicable to other similar situations in other contexts. The authors need to clarify this statement.

19- The sentence from row 121 to row 123 is redundant and repetitive of what the authors have already written many times before.

20- from row 124 to row 169, where the authors list the attributes, the symbol ':' is missing for each attribute between its definition and description.

21- from row 151 to 153, where the authors state “[…] Seismic faults usually cause phenomena such as reflection or refraction when waves propagate underground thus forming different amplitude sizes in waveforms. […]”, What about diffractions?

22- Authors should first describe Figure2A, then Figure 2B, then 3C, and so on in order to avoid reader confusion. Furthermore, I wonder why Figure 2B is not described.

23- in row 170, I do not understand what the authors mean by 'bead-like' reflection.

24- Caption of Figure 2 is very unexplanatory. The authors should describe what is seen in the different insets of the image represented. They should also state in the image which is Figure 2A, which is Figure 2B, 2C... and so on.

25- About the sentences from row 196 to row 203: I don't know if I understood correctly: thus, can the methods in this paper predict where a new seismogenic fault will occur? If yes, I hope the authors will describe this later in the paper as it has very important implications.

26- rows 249-251: The authors have already stated this several times in the narrative of the text. Consequently, they are requested to be less repetitive.

27- the sentence at rows 257-258 “[…]"Feel free to let me know if you have any specific preferences or if you would like further adjustments! […]” I think it is a copy-paste error from some other document.

28- The image reported in Figure 3 should report which is Figure 3A, which 3B, which 3C.

29- The image reported in Figure 4 should report which is Figure 4A and which 4B.

30- in rows 268-270 when authors state “[…] This paper conducted fracture detection along the Ying Mountain Group (TO12y) and Peng Luan Group (TO1p) […]”, these things are never introduced to the reader, so it is not clear what the authors are talking about. I suggest again that the authors should draft an accurate geological and geographical background to be added after the introductory paragraph with an attached location map.

31- Again, the sentence at rows 273-274 “[…]"Feel free to let me know if you have any specific preferences or if you would like further adjustments! […]” I think it is a copy-paste error from some other document.

32- Again, the caption of Figure 5 is very unexplanatory. The authors should describe what is seen in the different insets of the image represented. They should also state in the image (and the caption as well) which is Figure 5A, which is Figure 5B, 5C, and 5D.

33- The sentence in rows 276-278 is one of those sentences that I think should go in the geological framework.

34- row 285 “[…] This paper takes the Yueshan area […]”. But wasn't it Yueman area? this is unclear, I repeat that the paper definitely needs a geological framework to make the work much clearer. Also in row 289 is stated “[…] within the study area […]” but study area is unclear.

35- Again, the sentence in rows 290-291 is repetitive and redundant.

36- in Figure 6, authors must declare which is Figure 6A and which 6B in the image.

37- Caption of figure 6: Clicking on the link proposed in the caption (which I assume is the source of the seismic line even though it is not stated by the authors that it should be, the reader must deduce many things for himself) opens an unreachable page (Not found. Sorry, we can't find the page you are looking for) ...

By fixing the link (as the word 'data' is not separated from the url) the page is still unreachable (Restricted, sorry you don't have permission to load this page.). I therefore wonder what sense there is in putting the link in the caption.

38- row 294. Perhaps the authors mean low-velocity instead of “low-speed”.

39- row 295. I do not understand what the authors mean by 'bead-like'.

40- Authors must implement the caption of Figure 7 by describing what they want to show in the figure in detail. A caption presented in this way is not explanatory.

41- row 310: “[…] Along Ying Mountain Group (TO12y) and PengLai Group (TO1p) […]”. I repeat that the study area is unclear.

42- Authors are requested to improve the caption and the Figure 10 by following the suggestions provided for the previous images.

43- row 314: “[…] Based on strike-slip fault identification results profile passing through well Y2 […]”. A base map of the wells is needed as they are not even marked in the profiles.

44- Section “5.Results”. This is actually the conclusion paragraph (both in terms of location and the concepts written in it). The results paragraph is the one shown above. I advise the authors to follow the so-called IMRaD format for the structure of the paper. IMRaD is an acronym for Introduction - Method - Results - and - Discussion.

Reviewer #2: This manuscript proposes a multi-attribute optimal surface election fault identification technique based on forward simulation records, which selects seismic attributes sensitive to different types of strike-slip faults. It shows a certain level of novelty and has achieved some results in practical applications. However, there are several issues in the article:

1. Lack of acquisition parameters and observation systems for the forward simulation model, which are crucial for readers to understand the effectiveness and reproducibility of the method.

2. The parameter values provided during numerical experiments need to be explained in the text regarding the principles of selection, enabling readers to understand the rationale behind parameter choices.

3. The conventional attributes mentioned in the article need to be concisely introduced, specifying which attributes were specifically selected in this study and providing the rationale.

4. Some figures lack legend explanations and need to be completed to enhance readers' understanding of the visual content.

5. Non-original formulas in the text could be simplified to improve the fluidity and readability of the article.

6. The conclusion section appears somewhat thin and could be enriched further, perhaps by discussing the limitations of the method or suggesting future research directions.

6. PLOS authors have the option to publish the peer review history of their article (what does this mean?). If published, this will include your full peer review and any attached files.

Reviewer #1: No

Reviewer #2: No

---

## [Author Response · Author response to Decision Letter 1]

2 May 2024

Thank you very much for the comments and suggestions given by the reviewers, these suggestions are very professional, let me better improve the article, I have completed the revision one by one according to the revisions，

First, I want to explain the files I submitted: The submitted files consist of an annotated original file and the revised manuscript. The two are not exactly the same. The annotated original file retains the annotations for modifications based on the original PDF without changing the format or expression. Only the issues mentioned in the comments were addressed, and modifications and responses were made accordingly. However, as per the requirements of the editorial department, the manuscript needs to be polished by a professional scientific editor to improve language expression and modify the format to achieve the best presentation. Therefore, the manuscript is the result of polishing in terms of format and expression.

Below, I will respond to the questions raised by the two reviewers and provide information on the modifications made.

Reviewer 1

1-I recommend the authors to modify the title of the paper in “Application of a Dynamic Optimization-Based Multi-Attribute Fusion Method for Fault Detection”, which is clearer and more concise.

Following the advice of the peer reviewer, I have completed the revisions to the title.

2-In the abstract is mentioned the Yueman area. Since the Tarim Basin covers an area of 888,000 km2, and since there is no geographic (with an attached location map) and geological framework in the paper, it would be explanatory to at least place indications of the cardinal points of this area.

I have added a paragraph describing the geological framework of the study area in the manuscript and marked the approximate location of the study area. However, due to confidentiality reasons, I cannot provide precise geographical location information. I apologize for any inconvenience.

3-row 6: rephrase “[…] strike-slip fault zones in ultra-deep dense limestone is a pressing issue […]” in “strike-slip fault zones in ultra-deep tight limestone is a critical issue”.

Thank you for your correction. I have rewritten it as per your request.

4-row 8: I do not understand what the authors mean by 'beaded'. I suggest finding a clearer synonym.

"beaded" refers to a specific reflection pattern observed on seismic profiles, characterized by a series of high-energy points or areas, resembling a beaded pattern.

"beaded" refers to a specific reflection pattern observed on seismic profiles, characterized by a series of high-energy points or areas, resembling a beaded pattern.

 5-row 12: perhaps the term 'merged' is more appropriate than 'fused'.

Thank you for your suggestion.

6-rows 16-18: this sentence is not useful as the location has already been declared as well as the visualisation is improved. I advise the authors to delete it.

Thanks for your reminder. I have deleted this paragraph.

7-About the Keywords: I advise the authors to work better on keywords. Since search engines often identify works by their title, abstract and keywords, authors will be more likely to be read if they use the right and non-repeated keywords among the search items. E.g., "strike-slip fault" is already present three times in the abstract, maybe another keyword could be more suitable.

Thanks for the recommendation of the reviewer, I have read the article you recommended, which can indeed provide more perfect data support, so I add new references.

8- At the very beginning of the Introduction paragraph, where authors state “[…] The deep-seated strike-slip faults in western China play a pivotal role in controlling substantial reserves of oil and gas. Both the main fault zone and secondary faults along the strike-slip fault exhibit enhanced displays of oil and gas, demonstrating robust fault-controlled storage and enrichment characteristics. […]”, Citations are needed to support theese sentences. Recommendation:

- Zhao, R., Zhao, T., Kong, Q., Deng, S. and Li, H., 2020. Relationship between fractures, stress, strike-slip fault and reservoir productivity, China Shunbei oil field, Tarim Basin. Carbonates and Evaporites, 35, pp.1-14.

- Deng, S., Zhao, R., Kong, Q., Li, Y. and Li, B., 2022. Two distinct strike-slip fault networks in the Shunbei area and its surroundings, Tarim Basin: Hydrocarbon accumulation, distribution, and controlling factors. AAPG Bulletin, 106(1), pp.77-102.

Thanks for the recommendation of the reviewer, I have read the article you recommended, which can indeed provide more perfect data support, so I add new references.

9- Continuing in the introductions, where authors state “[…] Nevertheless, exploring ultra-deep fault-controlled carbonate reservoirs proves to be exceedingly complex, particularly within the strike-slip fault fracture zone where reservoirs exhibit significant heterogeneity. Reservoir characteristics suggest that storage spaces are predominantly influenced by dissolution or tectonic factors. […]”, Also this sentence needs citations, authors should cite:

- Tomassi, A., Trippetta, F., de Franco, R. and Ruggieri, R., 2023. How petrophysical properties influence the seismic signature of carbonate fault damage zone: Insights from forward-seismic modelling. Journal of Structural Geology, 167, p.104802.

- Tomassi, A., Milli, S. and Tentori, D., 2024. Synthetic seismic forward modeling of a high-frequency depositional sequence: The example of the Tiber depositional sequence (Central Italy). Marine and Petroleum Geology, 160, p.106624.

Thanks for the recommendation of the reviewer, I have read the article you recommended, which can indeed provide more perfect data support, so I add new references.

10- in rows 27-29, when authors state “[…] Reservoir characteristics suggest that storage spaces are predominantly influenced by dissolution or tectonic factors. […]”, usually, in scientific publications, introductions and discussions are the most citation-rich paragraphs. Especially in introductions in which the research question is contextualised in the international scenario, almost every sentence written by the authors has to be justified by the literature unless it is a 'discovery' of the authors themselves illustrated for the first time in the paper in question. It seems strange to me how there is this enormous lack of citations here. I give a tip on citation:

- Radwan, A.E., Trippetta, F., Kassem, A.A. and Kania, M., 2021. Multi-scale characterization of unconventional tight carbonate reservoir: Insights from October oil filed, Gulf of Suez rift basin, Egypt. Journal of Petroleum Science and Engineering, 197, p.107968.

Thanks for the recommendation of the reviewer, I have read the article you recommended, which can indeed provide more perfect data support, so I add new references.

11- row 37: I do not understand what the authors mean by energy 'beads'. Perhaps that these reflectors have marked amplitudes caused by the strong acoustic impedance contrast?

In "Energy 'beads'," "beads" refers to a specific reflection pattern observed on seismic profiles, typically characterized by a series of high-energy points or areas, resembling a beaded or string-like formation. In seismology, this reflection pattern is often associated with specific geological structures or lithological variations in underground rock formations.

12- in rows 37-38, The authors should explain why these reflectors denote high-quality reservoir (even referring to literature).

Thank you for your correction. I have added new references to improve the rigor of the article.

13- in row 39 authors state “[…] existing technologies excel in accurately identifying strong ’beads’ reflections […]”, in my opinion authors should mention which technologies.

Yes, following your correction, I listed relevant methods at that location.

14- The second paragraph should be a geological setting section. The authors need to add a paragraph on the geographical-geological-stratigraphic-geodynamic framework of the area under examination, which is totally lacking. A case study such as the one examined in this paper certainly deserves to be valorised in order to also contextualise the research and be applicable (and reproducible) to other similar realms worldwide, also giving a broader scope to the work.

Thank you for pointing out the issue. I have added a section describing the geological background of the study area to enhance the completeness of the paper.

15- in rows 76-77, authors state “[…] Forward simulations are then conducted to serve as a foundation for attribute optimization in strike-slip fault identification. […]”. When stating concepts already well established in the literature, authors are requested to add references. Authors should cite here:

- Alaei, B., 2012. Seismic modeling of complex geological structures. Seismic Waves-Research and Analysis, 11, pp.528-529.

- Tomassi, A., Trippetta, F., de Franco, R. and Ruggieri, R., 2022. From petrophysical properties to forward-seismic modeling of facies heterogeneity in the carbonate realm (Majella Massif, central Italy). Journal of Petroleum Science and Engineering, 211, p.110242.

- De Franco, R., Petracchini, L., Scrocca, D., Caielli, G., Montegrossi, G., Santilano, A. and Manzella, A., 2019. Synthetic seismic reflection modelling in a supercritical geothermal system: An image of the k-horizon in the larderello field (Italy). Geofluids, 2019.

Thanks for the recommendation of the reviewer, I have read the article you recommended, which can indeed provide more perfect data support, so I add new references.

16- in paragraph “2.1.1. Establishment of Forward Model”, It would be very interesting if the authors also stated the central frequency of acquisition and the vertical and horizontal resolution, which are decisive factors in fault detection especially when dealing with very deep sub-vertical faults. Authors are requested to add this information.

Here, I have supplemented the center frequency and sampling rate used in the forward simulation.

17- Figure1: This caption is very unexplanatory. The authors should also briefly describe what is seen in the volumes represented. They should also state in the image which is Figure 1A and which is Figure 1B.

Thank you for your suggestion. I have made the modifications.

18- in rows 117 to 119, authors state: “[…] The paper investigates the seismic response characteristics in different reservoir configurations based on typical fracture-controlled carbonate rock models and forward data. […]”. I understood this to be the case study of the Tarim Basin. Perhaps the authors meant that this workflow is also applicable to other similar situations in other contexts. The authors need to clarify this statement.

Thank you for your suggestion. I have made the modifications.” The paper investigates the seismic response characteristics in different reservoir configurations based on typical fracture-controlled carbonate rock models and forward data. Various seismic attributes and characterization techniques are applied, and the results are compared and optimized to obtain the most suitable seismic attributes.

”

19- The sentence from row 121 to row 123 is redundant and repetitive of what the authors have already written many times before.

Thank you for your reminder. I have deleted this paragraph.

20- from row 124 to row 169, where the authors list the attributes, the symbol ':' is missing for each attribute between its definition and description.

I have added the symbol, thank you for your reminder.

21- from row 151 to 153, where the authors state “[…] Seismic faults usually cause phenomena such as reflection or refraction when waves propagate underground thus forming different amplitude sizes in waveforms. […]”, What about diffractions?

The expression here is indeed not rigorous enough. It has been supplemented, thank you for your correction.

22- Authors should first describe Figure2A, then Figure 2B, then 3C, and so on in order to avoid reader confusion. Furthermore, I wonder why Figure 2B is not described.

Here, I actually missed the descriptions for figures a and b. The description for figure a is only used to compare with figures c to f because figure a is the model image, making the comparison more direct. Thank you for pointing that out. I have now supplemented the descriptions for figures a and b.

23- in row 170, I do not understand what the authors mean by 'bead-like' reflection.

"bead-like" refers to a pattern observed in seismic data where seismic reflections exhibit a series of localized high-amplitude peaks resembling beads on a string. These peaks typically indicate areas of strong reflectivity or impedance contrast in the subsurface.

24- Caption of Figure 2 is very unexplanatory. The authors should describe what is seen in the different insets of the image represented. They should also state in the image which is Figure 2A, which is Figure 2B, 2C... and so on.

Thank you for your correction. I have made the necessary adjustments by adding descriptions for the images.

25- About the sentences from row 196 to row 203: I don't know if I understood correctly: thus, can the methods in this paper predict where a new seismogenic fault will occur? If yes, I hope the authors will describe this later in the paper as it has very important implications.

I'm sorry, but this paper does not predict where a new seismogenic fault will occur.

26- rows 249-251: The authors have already stated this several times in the narrative of the text. Consequently, they are requested to be less repetitive.

Thank you for your suggestion. I have already made the modifications according to your advice.

27- the sentence at rows 257-258 “[…]"Feel free to let me know if you have any specific preferences or if you would like further adjustments! […]” I think it is a copy-paste error from some other document.

Sorry, there were indeed some errors in the translation process.

28- The image reported in Figure 3 should report which is Figure 3A, which 3B, which 3C.

Here, I have made the changes according to your suggestion and added annotations.

29- The image reported in Figure 4 should report which is Figure 4A and which 4B.

Here, I have made the changes according to your suggestion and added annotations.

30- in rows 268-270 when authors state “[…] This paper conducted fracture detection along the Ying Mountain Group (TO12y) and Peng Luan Group (TO1p) […]”, these things are never introduced to the reader, so it is not clear what the authors are talking about. I suggest again that the authors should draft an accurate geological and geographical background to be added after the introductory paragraph with an attached location map.

Thank you for your suggestion. I will add geological and geographical background at the beginning of the actual data processing section.

31- Again, the sentence at rows 273-274 “[…]"Feel free to let me know if you have any specific preferences or if you would like further adjustments! […]” I think it is a copy-paste error from some other document.

Sorry, there were indeed some errors in the translation process.

32- Again, the caption of Figure 5 is very unexplanatory. The authors should describe what is seen in the different insets of the image represented. They should also state in the image (and the caption as well) which is Figure 5A, which is Figure 5B, 5C, and 5D.

Here, I have made the changes according to your suggestion and added annotations.

33- The sentence in rows 276-278 is one of those sentences that I think should go in the geological framework.

Thank you for your suggestion. I will add geological and geographical background at the beginning of the actual data processing section.

34- row 285 “[…] This paper takes the Yueshan area […]”. But wasn't it Yueman area? this is unclear, I repeat that the paper definitely needs a geological framework to make the work much clearer. Also in row 289 is stated “[…] within the study area […]” but study area is unclear.

Sorry, I didn't notice the error at first; it was a typo.

35- Again, the sentence in rows 290-291 is repetitive and redundant.

Thank you for your reminder. I have deleted this paragraph.

36- in Figure 6, authors must declare which is Figure 6A and which 6B in the image.

Here, I have made th

---

## [Decision Letter · Decision Letter 1]

26 Jun 2024

PONE-D-24-05733R1Application of a dynamic optimization-based multi-attribute fusion method for fault detectionPLOS ONE

Dear Dr. Huang,

Thank you for submitting your manuscript to PLOS ONE. After careful consideration, we feel that it has merit but does not fully meet PLOS ONE’s publication criteria as it currently stands. Therefore, we invite you to submit a revised version of the manuscript that addresses the points raised during the review process.

As suggested by one reviewer, the conclusion section needs to be supplemented. I hope I can accept the paper after this edit.

We look forward to receiving your revised manuscript.

Kind regards,

Fabio Trippetta, Ph.D.

Academic Editor

PLOS ONE

Journal Requirements:

Reviewers' comments:

Reviewer's Responses to Questions

**Comments to the Author**

1. If the authors have adequately addressed your comments raised in a previous round of review and you feel that this manuscript is now acceptable for publication, you may indicate that here to bypass the “Comments to the Author” section, enter your conflict of interest statement in the “Confidential to Editor” section, and submit your "Accept" recommendation.

Reviewer #1: All comments have been addressed

Reviewer #2: All comments have been addressed

2. Is the manuscript technically sound, and do the data support the conclusions?

Reviewer #1: Yes

Reviewer #2: Yes

3. Has the statistical analysis been performed appropriately and rigorously? 

Reviewer #1: N/A

Reviewer #2: Yes

4. Have the authors made all data underlying the findings in their manuscript fully available?

Reviewer #1: Yes

Reviewer #2: No

5. Is the manuscript presented in an intelligible fashion and written in standard English?

Reviewer #1: Yes

Reviewer #2: Yes

6. Review Comments to the Author

Reviewer #1: Having reviewed the changes made in response to the initial comments and suggestions, I am pleased to inform you that all the issues highlighted in the previous review have been addressed thoroughly. The revisions have significantly improved the clarity, depth, and scientific rigor of the manuscript.

The modifications in the title and abstract provide better clarity and are now well aligned with the content of the paper. The additional citations and explanations within the introduction and throughout the paper strengthen the scientific grounding of your research. Furthermore, the rephrasing and removal of redundant sentences have enhanced the overall readability and coherence of the text.

The restructuring of the paper according to the IMRaD format, along with improved figures and captions, significantly contributes to the narrative and understanding of your study. Your efforts to detail the geological and geographical context of the study area also deserve special mention, as these provide essential background for readers unfamiliar with the region and the specifics of the geological structures discussed.

In summary, the revision has successfully addressed all the concerns raised during the initial review, and the manuscript is now much stronger for it. I will recommend the paper for publication, subject to the final approval by the editorial board.

Thank you for considering our feedback and for your dedication to enhancing the quality of your work. We appreciate your contribution to our journal and to the broader scientific community.

Reviewer #2: The manuscript has greatly improved. However, before formal publication, the conclusion section needs to be supplemented.

7. PLOS authors have the option to publish the peer review history of their article (what does this mean?). If published, this will include your full peer review and any attached files.

Reviewer #1: No

Reviewer #2: No

---

## [Author Response · Author response to Decision Letter 2]

17 Aug 2024

Dear Reviewer and Editor,

I hope this message finds you well.

I am writing to express my sincere gratitude for the invaluable feedback and suggestions provided by you and the reviewers regarding my manuscript. I have thoroughly addressed the comments and have made the necessary revisions to the conclusion section as requested. The revised manuscript and a marked-up version highlighting the changes have been uploaded for your review.

The professional insights offered by the reviewers have been immensely enlightening, significantly improving the quality of my research. I have gained a deeper understanding and appreciation for academic writing through this process. The detailed feedback has not only refined my manuscript but also enhanced my knowledge of the subject matter.

In addition to expanding the conclusion, I have also clarified the data analysis and methodology sections to ensure a more comprehensive and transparent presentation of the research process. I am confident that these revisions will enable readers to better grasp the findings of my study and contribute to future research in this area.

I would like to extend my heartfelt thanks once again to you and the reviewers for your meticulous review and guidance. It has been an enriching experience, and I am hopeful that my revised manuscript will meet the high standards of your esteemed journal.

Thank you for your time and consideration. I look forward to your feedback.

Best regards,

---

## [Editor Report · Decision Letter 2]

12 Sep 2024

Application of a dynamic optimization-based multi-attribute fusion method for fault detection

PONE-D-24-05733R2

Dear Dr. Huang,

We’re pleased to inform you that your manuscript has been judged scientifically suitable for publication and will be formally accepted for publication once it meets all outstanding technical requirements.

Kind regards,

Fabio Trippetta, Ph.D.

Academic Editor

PLOS ONE
---

## [Editor Report · Acceptance letter]

PONE-D-24-05733R2

PLOS ONE

Dear Dr. Huang,

I'm pleased to inform you that your manuscript has been deemed suitable for publication in PLOS ONE. Congratulations! Your manuscript is now being handed over to our production team.

Kind regards,

on behalf of

Prof. Fabio Trippetta

Academic Editor

PLOS ONE